# Effects of Adding Pre-Fermented Fluid Prepared from Red Clover or Lucerne on Fermentation Quality and In Vitro Digestibility of Red Clover and Lucerne Silages

Lin Sun [1,†], Yun Jiang [1,2,†], Qinyin Ling [3], Na Na [1], Haiwen Xu [4], Diwakar Vyas [2], Adegbola Tolulope Adesogan [2] and Yanlin Xue [1,*]

1   Inner Mongolia Engineering Research Center of Development and Utilization of Microbial Resources in Silage, Inner Mongolia Academy of Agriculture and Animal Husbandry Science, Hohhot 010031, China; sunlin2013@126.com (L.S.); jiangyun0110@ufl.edu (Y.J.); 13684752695@163.com (N.N.)
2   Department of Animal Sciences, University of Florida, Gainesville, FL 32611, USA; diwakarvyas@ufl.edu (D.V.); adesogan@ufl.edu (A.T.A.)
3   Department of Biology, University of Florida, Gainesville, FL 32611, USA; lingqinyin@ufl.edu
4   College of Foreign Languages, Inner Mongolia University of Finance and Economics, Hohhot 010010, China; goods99@126.com
*   Correspondence: xueyanlin_1979@163.com; Tel.: +86-471-5295-628
†   These authors have contributed equally to this study and share first authorship.

**Abstract:** This study examined the effects of chopping or chopping + blender maceration of red clover or lucerne on pre-fermented juice (PFJ) and determined the effects of PFJs on the quality of red clover silage or lucerne silage. The PFJs from chopping red clover (PFJ-RC) or lucerne (PFJ-LC) had a higher lactic acid bacteria (LAB) count than that from chopping + blender maceration ($p < 0.05$) and were used as additives. Compared with the Control of both silages, adding PFJ increased LAB, lactic acid (LA), and in vitro digestibility of dry matter (*IV*DMD) ($p < 0.05$), while pH, acetic acid (AA), and ammonia nitrogen/total nitrogen ($NH_3$-N/TN) were decreased ($p < 0.05$). For red clover silages, the PFJ-RC treatment contained the greatest LAB and LA and the lowest pH and $NH_3$-N/TN among treatments ($p < 0.05$); similar results were observed in PFJ-LC treatment for lucerne silages ($p < 0.05$). The *IV*DMD of both silages correlated negatively with pH, AA, and $NH_3$-N/TN and positively with LA ($p < 0.05$). Overall, chopping alone was a better method for preparing PFJ. Adding PFJ at ensiling increased LA and decreased the pH, AA, and $NH_3$-N/TN of both silages. Ensiling lucerne or red clover with PFJ from the ensiling material had a more positive effect on the fermentation parameters mentioned above. Satisfactory fermentation parameters detected in the present study contributed to improving the *IV*DMD of both silages.

**Keywords:** pre-fermented juice; lucerne silage; red clover silage; fermentation quality; in vitro digestibility

## 1. Introduction

Red clover (*Trifolium pratense* L.) and lucerne (*Medicago sativa* L.) are important legume forages with high nutritional value and are difficult to ensile, especially at low DM contents, because of the low content of water-soluble carbohydrates (WSC) and high crude protein (CP) content and buffering capacity (BC) [1]. Inoculation with homofermentative lactic acid bacteria (LAB) can improve fermentation quality by accelerating and increasing the production of lactic acid (LA), thus achieving acidic conditions more rapidly [2–4]. However, inoculants may not effectively improve forage preservation if the availability of fermentation substrates are insufficient to permit LAB growth [5]. Pre-fermented juice (PFJ) is a fermentation juice made from fresh forage containing natural epiphytic microorganisms, which can subsequently be used as a silage inoculant. The addition of PFJ at ensiling was more effective at improving the fermentation quality than LAB inoculation in some previous studies [2] because epiphytic LAB from PFJ stimulates LA production more

effectively, especially in high-moisture legume silages, which are typically difficult to ensile [6,7].

Several authors have reported improvements in fermentation quality of high-moisture legume silages by adding PFJ [2,8]. Denek et al. [9] reported that adding PFJ prepared from barley, wheat and grass herbages improved the fermentation quality of lucerne silage. Similarly, PFJ from lucerne was regarded as a good source of LAB for ensiling lucerne, inhibiting proteolysis and production of non-protein nitrogen (NPN) [2]. Ohshima et al. [10] showed that PFJ prepared from ensiling material is more effective at improving the fermentation quality than that prepared from other forage sources, possibly due to the better survival and growth of epiphytic LAB during ensiling. However, to our knowledge, no studies have reported the difference in composition of lactic acid bacteria in PFJs made from various forages, which is critical to explain the mode of action of PFJ and can be useful to develop more effective inoculants for various forages.

Two PFJ preparation methods, chopping and chopping + blender maceration, are used in the silage processing industry. While chopping reduces forage particle length, chopping + blender macerating samples reduce it even further. The differences between these methods in the LAB count and quality of PFJ have not been studied previously. The first objective of this study was to evaluate the effects of chopping and chopping + blender maceration on the characteristics of PFJ prepared from red clover and lucerne. The second objective was to identify the LAB species isolated from the PFJs and determine fermentation quality of red clover silage and lucerne silage inoculated with PFJ made from lucerne or red clover. We hypothesized that the chopped PFJ would have better quality compared with the chopping + blender macerating PFJ. Additionally, we hypothesized that PFJ from the forage being ensiled will be more effective at improving fermentation than that from the alternative forage.

## 2. Materials and Methods

### 2.1. Preparing Pre-Fermented Juices

Red clover and lucerne were grown for 4 years on an experimental farm (40° 46.265′ N, 111° 39.851′ E) at the Inner Mongolia Academy of Agricultural and Animal Husbandry Science, Hohhot, China. The fresh forage was harvested from three locations in adjacent red clover and lucerne fields (200 m × 20 m), respectively, and used as replicates. Forages were harvested at the early bud stage of maturity on 1 June 2017 in the first cut. The sampling locations were located in the central line of the field and were as follows: 1 location was in central point, and 2 locations were 30 m from 2 short sides, respectively. The fresh forage from each location was separately chopped into 10 mm pieces using scissors, thoroughly mixed, and then randomly divided into 2 batches. The PFJ was prepared according to the method described by Wang et al. [2]. Briefly, 200 g chopped forage of per batch from each location was mixed with 1000 mL of distilled water in glass bottles to which glucose was added (at 2 g per 100 mL ($w/v$)) according to Denek et al. [9]. The resulting PFJs of chopped red clover and lucerne were designated PFJ-RC and PFJ-LC, respectively. The 200 g chopped forage of another batch from each location was macerated in 1000 mL of distilled water for 1 min in a blender (Kinematica VS-5000YJ, Wuxi Warfaith Instrument Co. Ltd., Wuxi, China) and filtered through two layers of cheesecloth. Subsequently, the filtrates were collected in glass bottles to which glucose (2 g per 100 mL ($w/v$)) was added. All the bottles were sealed with gas trap seals and incubated in an incubator (LRH-70, Shanghai Yiheng Science Instruments Co. Ltd., Shanghai, China) at 30 °C for 48 h.

### 2.2. LAB Strains, Genomic DNA Extraction and Species Identification

Lactic acid bacteria were isolated from the PFJ prepared from both forages based on a procedure described earlier [11]. Briefly, Gram-positive and catalase-negative isolated strains were purified twice by streaking on Man, Rogosa and Sharpe (MRS) agar [12]. The purified colonies were grown on MRS agar at 30 °C for 24 h and stored in MRS broth containing 10% glycerol at −80 °C for further analysis [13].

After thawing, each isolated strain was cultured in 5 mL of MRS broth at 30 °C for 24 h in the same incubator and then centrifuged at $10,000 \times g$ for 5 min (5427 R, Eppendorf, Hamburg, Germany). The LAB cells were washed twice using Tris-EDTA buffer (10 mmol $L^{-1}$ Tris-HCl, 0.1 mmol $L^{-1}$ EDTA, pH 8.0) and centrifuged again at $10,000 \times g$ for 5 min. Genomic DNA extraction of LAB strains and identification were carried out according to Chen et al. [14]. The genomic DNA was extracted with TIANamp Bacterial DNA kit (Tiangen Biotech Co., Ltd., Beijing, China) according to the manufacturers' instructions. The concentration of genomic DNA was measured at 260 nm using a NanoDrop2000 spectrophotometer (Thermo Fisher Scientific, Waltham, MA, USA). The 16S rRNA gene region was amplified by the polymerase chain reaction (PCR) using the following universal primer of the 16S rRNA gene: 27F (5′-AGAGTTTGATCCTGGCTCAG-3′) and 1492R (5′-GGTTACCTTGTTACGACTT-3′). Exactly 25 μL of reaction volume was used, which contained 1 μL of diluted DNA template (approximately 80 ng), $10 \times$ PCR buffer solution (0.1 M Tris-HCl, pH 8.0, 0.5 M KCl), 1.5 mM $MgCl_2$ (pH 8.0), 100 μM of each dNTP, 1 U of AmpliTaq DNA recombinant polymerase and 0.4 μM of each primer. The PCR thermocycler conditions used were: preheating at 94 °C for 4 min, 25 cycles of denaturation at 94 °C for 30 s, annealing at 50 °C for 20 s, extension at 60 °C for 3 min and final extension at 72 °C for 10 min.

Exactly 1 μL of the PCR reaction mixture was detected by 1.5% agarose gel electrophoresis in $1 \times$ Tris-borate-EDTA buffer. The DNA bands were visualized under UV light (ZF-288, Shanghai Jiapeng Technology Co., Ltd., Shanghai, China). The PCR products were purified with a DNA purification system (Promega, Madison, WI, USA) and then sequenced using a 3730xl DNA analyzer (Applied Biosystems, San Francisco, CA, USA). The 16S rRNA gene sequences of the isolated LAB strains were analyzed by BLAST (http://www.ncbi.nlm.nih.gov/BLAST (accessed on 2 April 2018)) using GenBank as the database. The sequence similarity between the isolated strain and the standard strain was more than 99%, which identified the name of the isolated strain.

### 2.3. Silage Preparation

Red clover and lucerne were harvested from the same three locations in the fields as the forage samples prepared for pre-fermented juices in the morning of 4 June 2017 in the first cut, wilted in the field, tedded every 2 h by hand for 8 h (the DM content was about 400 g/kg) and carried to laboratory. The wilted forages from each location were chopped to 10–20 mm lengths using a chaffcutter (Hongguang Industry & Trade Co. Ltd., Ningbo, China), thoroughly mixed, and then randomly divided into 3 batches for 3 treatments. The LAB counts in chopped red clover and chopped lucerne before ensiling were 4.15 and 4.00 (log CFU/mL FW), respectively. The treatments were as follows: Control, spraying 3.0 mL/kg (fresh weight, FW) distilled water; PFJ-RC, spraying 3.0 mL/kg (FW) of PFJ made from chopped red clover; PFJ-LC, spraying 3.0 mL/kg (FW) of PFJ made from chopped lucerne. After mixing uniformly, approximately 300 g of treated forage was packed into a plastic bag (food grade, 180 mm × 260 mm; Qingye, Beijing, China), which was subsequently sealed with a vacuum sealer (DZ-300; Qingye, Beijing, China). In North China, the silage of lucerne is generally opened to feed livestock after 45 to 60 d of ensiling; thus, the silage bags in the present study were stored in a dark room at room temperature for 60 days.

### 2.4. Analysis

The LAB in the PFJs were cultured on MRS agar plates and counted [15]. The pH of PFJs was measured with a pH meter (PB-10, Sartorius AG, Goettingen, Germany). The PFJs were stored at 4 °C in a refrigerator (HYC-390, Haier lnc., Qingdao, China).

Silage bags were opened after 60 d of ensiling. Dry matter content was determined by drying samples in a forced-air oven at 65 °C for 48 h. Dry samples were ground to 1 mm using a mill (FS-6D; Fichi Machinery Qquipment Co. Ltd., Jinan, China) and then stored for further analysis. The total nitrogen (TN) was measured by the Kjeldahl method

by an autoanalyzer (Kjeltec 8400; FOSS Co. Ltd., Hillerød, Danmark) using copper as a catalyst and the crude protein (CP) concentration was calculated by multiplying the TN concentration by 6.25. Water-soluble carbohydrates were determined according to the method of McDonald and Henderson [16] using a spectrophotometer (Genesys 10; Thermo Fisher Scientific, Waltham, MA, USA). Neutral detergent fiber (NDF) and acid detergent fiber (ADF) were determined as described by Van Soest et al. [17] using an Ankom 2000 fiber analyzer (Ankom, Macedon, NY, USA) without using a heat stable amylase and expressed inclusive of ash. Acid detergent lignin (ADL) was measured using 72% $H_2SO_4$ solution using the method adapted by Van Soest et al. [17]. The buffering capacity was determined according to Playne and McDonald [18].

In vitro DM digestibility (*IV*DMD) was measured using a two-stage procedure [19] that first involved incubation of substrates in rumen fluid from 6 goats fed lucerne hay and whole-plant corn silage followed by incubation with pepsin solution. The pepsin solution was prepared by dissolving 2.0 g of 1:12,000 pepsin (Xi'an Tongze Biological Technology Co. Ltd., Xi'an, China) in 850 mL distilled water with 100 mL of 1 N HCl, and the final volume was adjusted to 1 L with distilled water. The concentrations of CP, NDF and ADF in the indigestible residue were determined, and the in vitro NDF digestibility (*IV*NDFD) and in vitro ADF digestibility (*IV*ADFD) were estimated as described by Filya et al. [20].

Silage extracts were prepared by blending 20 g of silage with 180 mL of distilled water for 1 min using a blender and then filtering the suspension through 4 layers of cheesecloth according to Owens et al. [21]. The LAB count in silages were determined according to Cai [11] by culturing on MRS agar. Immediately after filtration, pH of the filtrate was measured using a pH meter. The concentrations of lactic acid (LA), acetic acid (AA), propionic acid (PA), and butyric acid (BA) were measured by high-performance liquid chromatography (HPLC) (DAD, 210 nm, SPD-20A, Shimadzu Co., Ltd., Kyoto, Japan) with a column (Shodex RS Pak KC-811, Showa Denko K.K., Kawasaki, Japan) using 3 mM $HClO_4$ as the mobile phase with 1.0 mL/min flow rate set at 50 °C [22]. The concentration of $NH_3$-N was detected with an autoanalyzer using an adaptation of the Kjeldahl method.

### 2.5. Statistical Analyses

The differences in pH and LAB count of chopped and chopped + macerated PFJ of red clover or lucerne were analyzed with 2 treatments and 3 repetitions using the general linear model (GLM) procedure of SAS (version 9.1.3; SAS Inst. Inc., Cary, NC, USA). For the ensiling experiments, the differences among treatments of each silage were analyzed with 3 treatments and 3 repetitions by the GLM procedure of SAS. The differences were compared via least significant differences, and significance was declared at $p \leq 0.05$. The Pearson correlations of in vitro digestibility with LAB count, fermentation quality, and nutrient composition of red clover silage and lucerne silage were analyzed using R version 3.6.1.

### 3. Results

#### 3.1. pH and LAB Count of PFJs

The pH of PFJs made from chopping was lower ($p < 0.05$) compared with that from chopping + blender maceration for lucerne but similar ($p > 0.05$) for red clover. In addition, the PFJ made from chopping contained greater LAB count than that from chopping + blender maceration for both red clover and lucerne ($p < 0.05$) (Table 1). The PFJ-RC and PFJ-LC were used as additives.

#### 3.2. Identification of LAB Isolated from PFJs

There were 10 isolates including 5 genera and 7 species of LAB identified for PFJ-RC and 10 isolates including 6 genera and 8 species identified for PFJ-LC. The common species identified for both PFJ-RC and PFJ-LC were *Lactiplantibacillus plantarum*, *Leuconostoc mesenteroides*, *Enterococcus mundtii*, *Levilactobacillus brevis*, *Pediococcus pentosaceus*, and *Lactococcus*

*lactis* subsp. *cremoris*, while *Lacticaseibacillus paracasei* was unique for PFJ-RC. *Pediococcus acidilactici* and *Weissella cibaria* were only present in PFJ-LC (Table 2).

**Table 1.** pH and lactic acid bacteria (LAB, log CFU/mL FW) count of pre-fermented juice made from chopped, or blender macerated red clover or lucerne (n = 3).

| Items | | pH | LAB |
|---|---|---|---|
| Red clover | Chopping | 3.84 | 7.80a |
| | Chopping + blender maceration | 3.89 | 6.58b |
| | SEM | 0.062 | 0.069 |
| | *p*-value | 0.600 | <0.001 |
| Lucerne | Chopping | 3.77b | 7.57a |
| | Chopping + blender maceration | 5.11a | 6.54b |
| | SEM | 0.056 | 0.200 |
| | *p*-value | <0.001 | <0.001 |

Means within a column without common superscripts differ ($p < 0.05$). SEM, standard error of the mean.

**Table 2.** Lactic acid bacteria species isolated from pre-fermented juice made from chopped red clover and lucerne and identified by 16s rRNA sequencing (n = 3).

| Species | No. of Strains | | Similarity | Accession |
|---|---|---|---|---|
| | PFJ-RC | PFJ-LC | | |
| *Lactiplantibacillus plantarum* | 2 | 2 | >99% | NZ_CP030105.1 |
| *Leuconostoc mesenteroides* | 2 | 2 | >99% | NZ_CP028251.1 |
| *Enterococcus mundtii* | 2 | 1 | >99% | NZ_CP018061.1 |
| *Levilactobacillus brevis* | 1 | 1 | >99% | NZ_LS483405.1 |
| *Pediococcus pentosaceus* | 1 | 1 | >99% | NC_008525.1 |
| *Lactococcus lactis* subsp. *cremoris* | 1 | 1 | >99% | NC_022369.1 |
| *Lacticaseibacillus paracasei* | 1 | – | >99% | NC_014334.2 |
| *Pediococcus acidilactici* | – | 1 | >99% | NZ_CP053421.1 |
| *Weissella cibaria* | – | 1 | >99% | NZ_CP027563.1 |
| Total | 10 | 10 | | |

PFJ-RC, pre-fermented juice made from chopped red clover; PFJ-LC, pre-fermented juice made from chopped lucerne.

### 3.3. Forage Characteristics before Ensiling

The characteristics of red clover and lucerne before ensiling are presented in Table 3.

**Table 3.** Lactic acid bacteria (LAB) count, chemical composition, and in vitro digestibility of red clover and lucerne prior to ensiling (n = 3).

| Items | Red Clover | Lucerne |
|---|---|---|
| LAB count (log CFU/g FW) | 4.15 | 4.00 |
| Dry matter (DM, g/kg) | 401 | 398 |
| Crude protein (g/kg DM) | 195 | 206 |
| Water-soluble carbohydrates (g/kg DM) | 55.6 | 57.3 |
| Neutral detergent fiber (NDF, g/kg DM) | 350 | 410 |
| Acid detergent fiber (ADF, g/kg DM) | 268 | 282 |
| Acid detergent lignin (g/kg DM) | 106 | 99 |
| Buffering capacity (mEq/kg DM) | 583 | 425 |
| In vitro DM digestibility (g/kg) | 574 | 619 |
| In vitro NDF digestibility (g/kg) | 304 | 334 |
| In vitro ADF digestibility (g/kg) | 332 | 299 |

### 3.4. Fermentation Quality of Red Clover Silage and Lucerne Silage

Applying PFJ reduced pH and concentrations of acetic, propionic and butyric acids, and $NH_3$-N/TN and increased LAB count, lactic acid content, and lactic acid/acetic acid

for both silages compared to the Controls ($p < 0.05$). There were no propionic and butyric acids detected in both silages treated with PFJ. For red clover silage, the FFJ-RC treatment had the greatest LAB count, lactic acid content, and lactic acid/acetic acid and the lowest pH and $NH_3$-N/TN ($p < 0.05$); additionally, the Control contained lower WSC than PFJ treatments ($p < 0.05$). Similarly, for lucerne silage, the PFJ-LC treatment contained the greatest LAB count, lactic acid content, and lactic acid/acetic acid and the lowest pH and $NH_3$-N/TN ($p < 0.05$) (Table 4).

**Table 4.** Lactic acid bacteria (LAB) count, fermentation quality, and water-soluble carbohydrates (WSC) of red clover silages and lucerne silages (n = 3).

| Items | Red Clover Silage | | | | | Lucerne Silage | | | | |
|---|---|---|---|---|---|---|---|---|---|---|
| | Control | PFJ-RC | PFJ-LC | SEM | *p*-value | Control | PFJ-RC | PFJ-LC | SEM | *p*-Value |
| LAB (log CFU/g FW) | 6.13c | 8.22a | 7.51b | 0.038 | <0.001 | 6.25c | 7.29b | 8.21a | 0.108 | <0.001 |
| pH | 5.31a | 4.63c | 4.70b | 0.017 | <0.001 | 5.43a | 4.67b | 4.32c | 0.014 | <0.001 |
| LA (g/kg DM) | 7.65c | 41.81a | 29.83b | 1.44 | <0.001 | 8.58c | 30.60b | 44.70a | 1.51 | <0.001 |
| AA (g/kg DM) | 14.39a | 5.50b | 6.06b | 0.771 | <0.001 | 12.78a | 6.17b | 3.90b | 0.813 | <0.001 |
| PA (g/kg DM) | 0.23 | ND | ND | 0.009 | <0.001 | 0.28a | ND | ND | 0.008 | <0.001 |
| BA (g/kg DM) | 0.14a | ND | ND | 0.007 | <0.001 | 0.24a | ND | ND | 0.009 | <0.001 |
| LA/AA | 0.54c | 7.70a | 4.95b | 0.467 | <0.001 | 0.69c | 5.01b | 11.82a | 0.946 | <0.001 |
| $NH_3$-N/TN (g/kg) | 106.4a | 38.3c | 45.8b | 1.61 | <0.001 | 106.3a | 43.1b | 30.6c | 0.552 | <0.001 |
| WSC (g/kg DM) | 20.5b | 24.1a | 23.9a | 0.714 | 0.022 | 20.5 | 22.5 | 22.6 | 0.611 | 0.090 |

Means within a row without common superscripts differ ($p < 0.05$); SEM, standard error of the mean; ND, not detected. PFJ-RC, pre-fermented juice made from chopped red clover; PFJ-LC, pre-fermented juice made from chopped lucerne. LA, lactic acid; AA, acetic acid; PA, propionic acid; BA, butyric acid; $NH_3$-N/TN, ammonia nitrogen/total nitrogen.

### 3.5. Nutrient Composition and In Vitro Digestibility of Red Clover Silage and Lucerne Silage

For both silages, treating PFJ decreased NDF concentration and increased *IV*DMD compared with Control ($p < 0.05$); moreover, the PFJ-LC treatment contained the lowest ADF content and had the greatest *IV*DMD among treatments ($p < 0.05$). For red clover silage, the PFJ-RC treatment had the highest DM and ADL concentrations ($p < 0.05$). For lucerne silage, the PFJ-RC had the highest DM concentration ($p < 0.05$; Table 5).

**Table 5.** Nutrient composition and in vitro digestibility of red clover silage and lucerne silage (n = 3).

| Items | Red Clover Silage | | | | | Lucerne Silage | | | | |
|---|---|---|---|---|---|---|---|---|---|---|
| | Control | PFJ-RC | PFJ-LC | SEM | *p*-Value | Control | PFJ-RC | PFJ-LC | SEM | *p*-Value |
| DM (g/kg) | 385b | 393a | 380b | 1.63 | 0.005 | 371c | 390a | 381b | 2.01 | 0.015 |
| CP (g/kg DM) | 201 | 200 | 203 | 2.35 | 0.736 | 222 | 229 | 229 | 3.22 | 0.296 |
| NDF (g/kg DM) | 348a | 338b | 330b | 2.77 | 0.011 | 401a | 389b | 374c | 2.94 | 0.002 |
| ADF (g/kg DM) | 240a | 236a | 220b | 2.60 | 0.004 | 264a | 259a | 239b | 2.36 | <0.001 |
| ADL (g/kg DM) | 73.2b | 79.6a | 74.0b | 1.46 | 0.042 | 87.6 | 89.2 | 77.1 | 4.94 | 0.247 |
| BC (mEq/kg DM) | 716 | 705 | 696 | 12.7 | 0.557 | 536 | 539 | 532 | 11.7 | 0.908 |
| *IV*DMD (g/kg) | 624c | 664b | 684a | 2.89 | <0.001 | 631c | 652b | 705a | 5.33 | <0.001 |
| *IV*NDFD (g/kg) | 513 | 547 | 550 | 10.8 | 0.498 | 539 | 576 | 549 | 16.0 | 0.315 |
| *IV*ADFD (g/kg) | 497 | 492 | 492 | 8.25 | 0.893 | 439 | 458 | 484 | 10.7 | 0.068 |

Means within a row without common superscripts differ ($p < 0.05$); SEM, standard error of the mean. PFJ-RC, pre-fermented juice made from chopped red clover; PFJ-LC, pre-fermented juice made from chopped lucerne. DM, dry matter; CP, crude protein; NDF, neutral detergent fiber; ADF, acid detergent fiber; ADL, acid detergent lignin; BC, buffering capacity; *IV*DMD, in vitro DM digestibility; *IV*NDFD, in vitro NDF digestibility; *IV*ADFD, in vitro ADF digestibility.

### 3.6. Correlation between In Vitro Digestibility and Quality of Red Clover Silage and Lucerne Silage

For both silages, the *IV*DMD had positive correlation with LAB count, lactic acid content, and lactic acid/acetic acid and correlated negatively with pH, concentrations of acetic, propionic, and butyric acids, $NH_3$-N/TN, NDF, and ADF ($p < 0.05$). For red clover silage, the *IV*DMD also correlated positively with WSC concentration ($p < 0.05$). For lucerne



silage, the *IV*ADFD correlated positively with LAB count, lactic acid, lactic acid/acetic acid, and CP and had a negative correlation with pH, acetic acid and $NH_3$-N/TN ($p < 0.05$; Table 6).

**Table 6.** Pearson correlation of in vitro digestibility with lactic acid bacteria, fermentation quality, and nutrient composition of red clover silage and lucerne silage (n = 9).

| Items | Red Clover Silage | | | Lucerne Silage | | |
|---|---|---|---|---|---|---|
| | *IV*DMD | *IV*NDFD | *IV*ADFD | *IV*DMD | *IV*NDFD | *IV*ADFD |
| Lactic acid bacteria | 0.767 * | 0.433 | −0.176 | 0.894 ** | 0.186 | 0.784 * |
| pH | −0.899 *** | −0.439 | 0.167 | −0.875 ** | −0.256 | −0.722 * |
| Lactic acid | 0.781 * | 0.443 | −0.251 | 0.887 ** | 0.276 | 0.763 * |
| Acetic acid | −0.873 ** | −0.334 | 0.213 | −0.793 * | −0.381 | −0.746 * |
| Propionic acid | −0.932 *** | −0.515 | 0.172 | −0.696 * | −0.407 | −0.648 |
| Butyric acid | −0.920 *** | −0.39 | 0.215 | −0.692 * | −0.402 | −0.611 |
| Lactic acid/acetic acid | 0.741 * | 0.398 | −0.178 | 0.931 *** | 0.265 | 0.746 * |
| Ammonia nitrogen/total nitrogen | −0.886 ** | −0.428 | 0.170 | −0.796 * | −0.319 | −0.696 * |
| Water-soluble carbohydrates | 0.738 * | 0.209 | 0.014 | 0.530 | 0.329 | 0.484 |
| Dry matter | −0.179 | 0.226 | 0.133 | 0.347 | 0.601 | 0.297 |
| Crude protein | 0.074 | 0.084 | 0.567 | 0.289 | 0.357 | 0.732 * |
| Neutral detergent fiber | −0.933 *** | −0.530 | 0.325 | −0.951 *** | −0.192 | −0.524 |
| Acid detergent fiber | −0.788 * | −0.246 | 0.146 | −0.929 *** | −0.047 | −0.620 |
| Acid detergent lignin | 0.234 | 0.127 | −0.313 | −0.54 | 0.012 | −0.250 |
| Buffering capacity | −0.389 | 0.368 | 0.026 | −0.174 | −0.130 | −0.081 |

*IV*DMD, in vitro digestibility of dry matter; *IV*NDFD, in vitro digestibility of neutral detergent fiber; *IV*ADFD, in vitro digestibility of acid detergent fiber. *, $p < 0.05$; **, $p < 0.01$; ***, $p < 0.001$.

## 4. Discussion

In the current study, the glucose was added to the PFJ to act as a substrate that would promote a good fermentation in the presence of an adequate number of LAB. Supplementing WSC to PFJ for facilitating fermentation was previously suggested, and glucose was regarded as the best sugar substrate for the preservation of silage [5,10]. The higher LAB count of PFJs prepared from chopping than chopping + blender maceration indicated that chopping is a better processing method for preparing PFJ (Table 1). Consequently, the PFJ-RC and PFJ-LC were selected as silage additives in the present study. The LAB counts in the PFJs (7.57 and 7.80 log CFU/mL in PFJ-RC and PFJ-LC, respectively) were slightly lower than those reported by Ohshima et al. [6,7,10]. However, spraying 3.0 mL/kg (FW) of PFJ on materials in the present study were still adequate to achieve the LAB count ($10^5$ CFU/g FW) required for an adequate fermentation [1].

In the study of Tao et al. [23], only two genera of LAB were reported in PFJ prepared from lucerne. However, in the present study, seven and eight genera were isolated and identified from PFJ-RC and PFJ-LC, respectively. The differences between both studies might be attributed to different laboratory methods used to identify LAB. While Tao et al. [23] used selective agar media-based culturing, 16S rRNA gene sequencing, which has better coverage, was used in the present study. Six common LAB species were identified in PFJ-RC and PFJ-LC, probably because red clover and lucerne were harvested from the same experimental farm. Previous studies have detected a variety of LAB species in silages including *L. plantarum*, *P. pentosaceus*, *L. brevis*, *L. mesenteroides* and *L. lactis* [24–26]. In our study, all of these species were isolated from both PFJ-RC and PFJ-LC and accounted for over 60% of all identified LAB species. In addition, *L. paracasei* was identified only in PFJ-RC, while *W. cibaria* and *P. acidilactici* were only found in PFJ-LC (Table 2). The wide range of LAB species present in PFJs and high LAB count indicated the suitability of using PFJs as silage inoculants to facilitate fermentation. In addition, the different LAB composition of PFJ-RC and PFJ-LC indicated that red clover and lucerne contain different epiphytic LAB species.

Due to the low LAB count and WSC content and the higher buffering capacity of red clover and lucerne before ensiling (Table 3), it is difficult to achieve an efficient fermentation during ensiling [21]. This was reflected by the low lactic acid concentration and high pH and $NH_3$-N/TN in Control silages in the present study (Table 4). The propionic and butyric acids were detected only in Control silages at a very low level (Table 4) because the high DM content (401 and 398 g/kg, respectively) and ideal anaerobic environment may have caused a certain decrease in activity of undesired microorganisms during ensiling process in Control silages. The ratios of lactic acid to acetic acid in Control silages were below one (Table 4) and indicated that abnormal fermentations had occurred in the silages [27,28], resulting from activity of *Enterobacteriaceae* dominating the bacterial community in Control silages during fermentation process. *Enterobacteriaceae* could thrive in anaerobic and weak acidic conditions [29] and ferment WSC and lactic acid to acetic acid, succinic acids, ethanol, or 2,3-butanediol [30,31]. The presence of *Enterobacteriaceae* may have accounted for the low levels of WSC (20.5 and 20.5 g/kg DM, respectively), the low lactic acid (7.56 and 8.58 g/kg DM, respectively) and the high pH (5.31 and 5.43, respectively) in Controls of both silages (Table 4). Based on these above mentioned results, it is necessary to ensile red clover and lucerne with inoculants. However, as the valeric acid, caproic acid, ethanol, propanol, 1,2-propandiol, and DM losses were not analyzed in the present study, the effect of inoculating PFJ on fermentation quality of silages could not be evaluated accurately. Thus, additional analyses of the alcohols, acids, and DM losses stated above would have helped in better understand the effect of PFJ; this needs further study to fully evaluate the influence of ensiling red clover and lucerne with PFJ.

Successful silage fermentation mainly depends on the ensiling technique, properties of the inoculant, characteristics of plants ensiled, epiphytic microflora, and climatic conditions [32]. Ensiling legumes, which have a higher buffering capacity, with LAB can improve the fermentation quality and inhibit undesirable microorganisms [33]. According to the evaluation system for fermentation quality of silages based on the contents of butyric and acetic acids [34], the scores of all treatments in both silages were 100, and their mark was the highest (first) in the present study, because of the lower contents of butyric and acetic acids in all silage (<3.0 and <30 g/kg DM, respectively). The higher lactic-to-acetic ratio, LAB count, and lactic acid concentration and lower pH for PFJ treatments suggested homolactic fermentation dominated the ensiling process [28]. Our results agreed with Filya et al. [20] who showed that a number of commercial homolactic inoculants improved lucerne silage fermentation by shifting fermentation towards lactic acid and reducing the pH. In addition, Ohshima et al. [10] reported that ensiling lucerne and Italian ryegrass with PFJ containing high concentrations ($10^8$ CFU/mL) of epiphytic LAB increased lactic acid concentration and accelerated the decrease in silage pH. Our results are also in agreement with other studies [2,9,23], showing efficacy of PFJ at increasing LAB count and lactic acid and decreased pH, acetic acid, and $NH_3$-N/TN of silage (Table 4). According to the typical suggested concentrations of common fermentation end products in legume silage [28], in the present study, the PFJ treatments contained appropriate pH and lactic acid contents and had very low acetic acid in both silages, which resulted in a high lactic-to-acetic acid ratio in PFJ treatments.

The LAB of PFJ prepared from ensiling crops may grow better on the same forage during fermentation than the PFJ prepared from another crop [10]. In the present study, the forages for preparing PFJ and ensiling were grown in the same field. Ensiling red clover with PFJ-RC reduced pH and $NH_3$-N/TN and increased LAB count, lactic acid concentration, and lactic-to-acetic acid ratio to a greater extent than PFJ-LC (Table 4). Similarly, the PFJ-LC had more positive effect on the LAB, pH, lactic acid, and $NH_3$-N/TN of lucerne silages than PFJ-RC (Table 4). The results suggested that inoculating lucerne or red clover with PFJ prepared from the ensiling material was more effective at improving fermentation parameters detected in the present study. This might be due to the materials prepared for PFJ and silage being collected from the same field, and the LABs in PFJ are more familiar with the physico-chemical properties of the silage [35]. Similarly, Ali et al. [36]

revealed that the microbiota from red clover had a better effect on bacterial community succession and fermentation quality in red clover silage than that from maize and sorghum. In addition, Sun et al. [35] also found that the LAB from whole-plant corn silage had a better promotion effect on microbial succession and fermentation of whole-plant corn silage than those from *Elymus sibiricus* silage.

The $NH_3$-N is part of the non-protein in silage and $NH_3$-N/TN indicates the degree of silage preservation during fermentation [37,38]. In the present study, although the Control silages had the suggested contents of $NH_3$-N/TN according to Kung et al. [28], the PFJ treatments contained lower levels of $NH_3$-N/TN (Table 3). This indicated that applying PFJs decreased $NH_3$-N/TN in both red clover silages and lucerne silages, because adding PFJ could effectively reduce activity of *Enterobacteriaceae*. *Enterobacteriaceae* are responsible for much of the $NH_3$-N formed from protein degradation and from the reduction of $NO_3$ [32] in Control silages. This is in agreement with Wang et al. [2] and Tao et al. [23], who reported that lucerne silage treated with PFJ contained less $NH_3$-N/TN than the silage with commercial LAB additives in some cases. For both forages from the same experimental farm, the PFJ made from the ensiling material was more effective in well preserving silage than PFJ made from the other forage source.

Both PFJ treatments increased the *IV*DMD of both silages, but PFJ-LC increased the *IV*DMD to greater extent than PFJ-RC (Table 5). The increased *IV*DMD by ensiling red clover or lucerne with PFJs is consistent with Nishino and Uchida [8], who reported that adding PFJ to lucerne at ensiling increased the *IV*DMD. Similarly, Ellis et al. [39] also reported that LAB inoculation can improve the digestibility of silage. The increased *IV*DMD might be due to the satisfactory fermentation parameters detected in PFJ-treated silages contributing to the preservation of digestible nutrients in the present study, which were reflected by the *IV*DMD correlating positively with lactic acid and lactic acid/acetic acid and negatively with pH, acetic acid, and $NH_3$-N/TN (Table 6). Moreover, the plant cell wall negatively influences the digestibility of silage [40], which was reflected by the negative correlation of *IV*DMD with NDF and ADF for both silages in the present study (Table 6). Similarly, previous studies also reported that the higher *IV*DMD was detected in silages with lower NDF and ADF concentrations [27,41].

## 5. Conclusions

The processing of chopping produced PFJ with a greater LAB count than blender chopping + blender maceration. A total of seven and eight LAB species were isolated and identified from PFJ prepared from red clover and lucerne, respectively, with *P. acidilactici* and *W. cibaria* only present in PFJ made from lucerne, while *L. paracasei* was uniquely present in PFJ made from red clover. The forage for preparing PFJ and ensiling was collected from the same field, ensiling red clover or lucerne with PFJ made from the ensiling material had a more positive effect on pH, LA, AA, and $NH_3$-N/TN than PFJ made from the other forage sources. Both PFJ treatments improved the *IV*DMD of the lucerne and red clover silages. Satisfactory fermentation parameters detected in the present study helped to improve the *IV*DMD of both silages. The effect of inoculating PFJs prepared from different fields on the silage quality needs further research in the future.

**Author Contributions:** Conceptualization, L.S., Y.J., and Y.X.; methodology, L.S. and H.X.; software, L.S. and Q.L.; validation, Y.J. and Q.L.; formal analysis, L.S. and Y.J.; investigation, N.N., and Y.X.; resources, L.S. and Y.J.; data curation, Q.L., H.X., and N.N.; writing—original draft preparation, L.S. and Y.J.; writing—review and editing, Y.J., H.X., D.V., A.T.A., and Y.X.; supervision, Y.X.; project administration, Y.X.; funding acquisition, Y.X. All authors have read and agreed to the published version of the manuscript.

**Funding:** This study was funded by the National Key R&D Program of China (grant number, 2017YFE 0104300) and the Science and Technology Project of Inner Mongolia (grant number, 2020GG0049).

**Institutional Review Board Statement:** Not applicable.

**Informed Consent Statement:** Informed consent was obtained from all subjects involved in the study.

**Data Availability Statement:** The data presented in this study are available on request from the corresponding author.

**Conflicts of Interest:** The authors declare no conflict of interest.

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
