# Peer review of "Effects of Adding Pre-Fermented Fluid Prepared from Red Clover or Lucerne on Fermentation Quality and In Vitro Digestibility of Red Clover and Lucerne Silages"

_agriculture, doi:10.3390/agriculture11050454_

Round 1

Reviewer 1 Report

After correcting the manuscript may be published. 

Author Response

Dear Reviewer 1,

Your previous comments will be of great help to my future research.

Thanks!

All the best,

Yanlin Xue

Reviewer 2 Report

Throughout your discussion, when describing the benefits of using PFJ from the same crop, clearly state that PFJ was prepared with material from the same field. Your trial does not allow you to fully make inferences of crops.

L. 22 - Control

L. 47 - from PFJ stimulates LA

L. 62-63 - While chopping reduces forage particle length, blender macerating samples reduce it even further.

L. 76 - here and throughout - please replace district with location

L. 83 and 87 - forage

L. 131 - laboratory.

L. 134-136 - again, adding the inoculation rate as cfu/g of fresh forage is a requirement for inoculation studies and adds clarity to the manuscript and not ambiguity. 

L. 144 - delete space between 4 and ℃

L. 179 and 181 - did you really use this GLIM procedure? Or was this a typo for GLIMMIX?

L. 200 - (n = 3)

Table 4 - please revise table to ensure decimal places are the same for each treatment mean within a variable. 

L. 246 - (n = 3)

L. 315-320 - This is probably related to PFJ being from the same field and therefore having similar bacterial population and not only because of crop. 

L. 334-339 - confuse, please rewrite.

Conclusion -  Please add a statement suggesting that future research should be conducted with PFJ from different fields.

Author Response

Dear Reviewer 2,

Thanks for your suggestions about the manuscript.

Those comments will help me a lot in the future.

The attachment is the response to your comments. Please check it.

Thanks again.

All the best,

Yanlin Xue

This manuscript is a resubmission of an earlier submission. The following is a list of the peer review reports and author responses from that submission.

Round 1

Reviewer 1 Report

In generally this manuscript is very interesting. It was prepared well.

The manuscript have had an interesting concept and it is well prepared in terms of methodology. The manuscript have had a corretly methods but some analyzes of the juice and silage are missing, which would allowe for the better discussion.

It have an interesting concept, however it will be difficult to aplly in commercial conditions.

Please describe the role of these plants in agriculture. Please also extend the work to publications on additives for fermentation / prefermentation or during plant vegetation:

- Biostimulating effect of l-tryptophan on the yield and chemical and microbiological quality of perennial ryegrass (Lolium perenne) herbage and silage for ruminant

doi: 10.1002/jsfa.10999. Online ahead of print.

- Effects of additives on the fermentation and aerobic stability of grass silages and total mixed rations

https://doi.org/10.1111/gfs.12221

- The effect of additives on the quality of white lupin–wheat silage assessed by fermentation pattern and qPCR quantification of clostridia

https://doi.org/10.1111/gfs.12276

  • L 109-114: Please check and edit this paragraph. Wrong spelling.
  • L 180-185: It is an unnecessary cited of values from the adjacent table.
  • L 194: It would be worth analyse WSC content and CP in fresh PFj in order to explain to difference in pH.
  • L 199 (table 2): The whole layout of the tables on this page is bad. Please arrange all tables in the text, not on one page. Table 2 - Perhaps a vertical layout will be better?
  • L 202 (and materials and methods): Please provide the accession numbers and % sequence similarity of the 16S rRNA (BLAST/NCBI).
  • L 253: It is a worse numer of table.
  • L 254: In my opinion the WSC content would be help in interpretation.
  • L 296-297: It should be supported by a citated.
  • L 323: The N-NH3 content in herbage would be help.
  • L 325: On the based on N-NH3 content you must not talk about proteolysis, because interpretation of proteolisys need to NPN content. The factory that acted was the limitation of Clostridium by efficient lactic fermentation and it indicates that no butyric acid that is high corelated with N-NH3.
  • L 326: The analyses of NPN in fresh herbage, wilted herbage and silage would be give answer about range of proteolysis. I suggested you to cited study of Fijałkowska et al. 2015 (Fijałkowska M., Lipiński K., Pysera B., Wierzbowska J, Antoszkiewicz Z., Sienkiewicz S., Stasiewicz M. 2015. The effect of ensiling in round bales on the content of nitrogen fractions in lucerne and red clover protein. J. Elem., 20(2): 285-291. DOI: 10.5601/jelem.2014.19.4.643). These studies describe the factors that influence the proteolytic transformation.
  • Quite good discussion. Nevertheless the study lacks, for example: the correlation of various factors (eg: LAB vs. lactic acid), and there is no comprehensive description and comparison with other studies. PFJ - LAB additive - beneficial and undesirable chemical effects of silage.

Author Response

Dear reviewer,

Thank you for your valiable suggestions, which help me a lot in improving the manuscript.

The attacment is my response to your comments.

Thanks again.

All the best,

Yanlin Xue

Reviewer 2 Report

This is a well-designed study with potential implications to silage growers.

My main comment is related to your discussion. Although you briefly mention it, a more elaborated discussion of propionic and butyric acids, pH, and NH3-N for control would be of benefit to readers. There results are very suggestive a clostridial fermentation. 

Also, it looks like PFJ and ensiled forages were from the same field. This is an important information as epiphytic bacterial population was likely very similar and explain results observed for the best PFJ when ensiled these crops. Although you mentioned that using the same crop is advised, which I agree, it may also be related to the same field. 

L. 4 - delete "the"

L. 34 - carbohydrates

L. 36 - more rapidly

L. 37 - change to "may not effectively improve forage preservation"

L. 38 - change to "are not sufficient to permit LAB growth"

L. 42 - delete "they can"

L. 45 - change to "reported improvements on fermentation"

L. 46 - delete "was improved"

L. 58 - finer particles

L. 64 - would have better quality

L. 90 - add a space between containing and 10%

L. 94 - using Tris-EDTA buffer

L. 110 - 1 × Tris-borate-EDTA buffer.

L. 116 - Were red clover and alfalfa harvested from the same field as the material used to prepare PFJ? Please describe it here.

L. 121 to 124 - Please provide cfu of LAB/g of fresh forage used for each treatment.

L. 179 - LAB count of 4.15 log CFU/g of FW

Table 1 - For each variable, use the same decimal places between both forages (see Hemicellulose)

L. 216 - and butyric acids as well

L. 220 -  and butyric acids as well

Table 4 - same as comment for Table 1

L. 265 - please provide specific dose used for this study in the M&M section and here 

L. 274 - Lactobacillus

L. 284  to 288 - I would suggest you to rewrite this sentence and state that propionic and butyric acids were detected only for Control samples instead of mentioning low concentrations of butyric acid.

L. 306 - Did you collect samples for PFJ preparation and material for ensiling from the same field? If yes, please mention it here. 

L. 308 - delete significant it is redundant

L. 319 -add the forage instead of saying other forage

L. 335-336 - I don't agree with this explanation as alfalfa and red clover have negligible concentration of starch. WSC content after ensiling may explain this effect for red clover.

L. 344 - greater LAB counts instead of quality 

Author Response

(The authors gave the same response as above.)
